# Atomistic Insights into the Phase Transformation of Single-Crystal Silicon during Nanoindentation

**DOI:** 10.3390/nano12122071

**Published:** 2022-06-16

**Authors:** Young Jin Chung, Gi Hun Lee, Hyeon Gyu Beom

**Affiliations:** Department of Mechanical Engineering, Inha University, 100 Inha-ro, Incheon 22212, Korea; dudwls6426@naver.com (Y.J.C.); solafide27@gmail.com (G.H.L.)

**Keywords:** nanoindentation, indenter angle, phase transformation, silicon, atomistic simulation

## Abstract

The influence of the indenter angle on the deformation mechanisms of single-crystal Si was analyzed via molecular dynamics simulations of the nanoindentation process. Three different types of diamond conical indenters with semi-angles of 45°, 60°, and 70° were used. The load–indentation depth curves were obtained by varying the indenter angles, and the structural phase transformations of single-crystal Si were observed from an atomistic view. In addition, the hardness and elastic modulus with varying indenter angles were evaluated based on the Oliver–Pharr method and Sneddon’s solution. The simulation results showed that the indenter angle had a significant effect on the load–indentation depth curves, which resulted from the strong dependence of the elastic and plastic deformation ratios on the indenter angle during indentations.

## 1. Introduction

The highly localized stress near the indenter tip induces characteristic plastic responses in the material [1]. For instance, thin films consisting of single-crystal Si beneath a rigid indenter exhibit phase transformations during indentation [2,3]. When the stress in single-crystal Si with a diamond cubic structure (Si-I) increases to nearly 12 GPa, the cubic diamond phase is transformed to a β-tin structure (Si-II) [4]. For the unloading procedure, metastable phases such as body-centered cubic (Si-III) or rhombohedral distortion (Si-XII) can occur [5,6]. This phase transformation in single-crystal Si during indentation has been found to be an important factor that determines its mechanical properties. However, Jang et al. [1] investigated triangular pyramidal indenters with four different indenter angles and experimentally demonstrated that the transformation behavior of Si was strongly dependent on the shape of the indenter.

Micro-Raman analysis and transmission electron microscopy (TEM) are too restricted to provide sufficient explanations for the sequence of phase transformations [7]. An atomistic analysis via molecular dynamics (MD) simulations can effectively elucidate the essential features of phase transformations, compared to the limited observations from experiments [8,9,10]. Recently, several atomistic studies on the nanoindentation of thin-film Si have been conducted by Sun et al. [11], Wang et al. [12], and Jiao et al. [13]. However, these computational studies focused on the use of a spherical indenter. Different geometries of the indenter tip significantly influence the stresses and strains of the material beneath the indenter, which may induce deviations in the mechanical properties from theoretical values [14,15].

The present study investigates the effect of the indenter angle on the indentation behavior of single-crystal Si using MD simulations. Nanoindentations were performed using a perfect conical indenter, and the indenter semi-angles were chosen as 45°, 60°, and 70°. MD simulations were conducted at temperatures of 1 K and 300 K. The relationship between the indenter angle and the deformation mechanism of single-crystal Si was analyzed through load–indentation depth (*P*-*h*) curves and atomistic views. In addition, mechanical properties such as hardness and elastic modulus were evaluated using the *P*-*h* curve, as suggested by Oliver and Pharr [16] and Sneddon [17].

## 2. MD Simulation Method

The atomistic model used in this study is shown in Figure 1. The *x-*, *y-*, and *z*-axes of the model are oriented along the [100], [010], and [001] crystallographic directions, respectively. The dimensions of the Si substrate were chosen as *L*_x_ = 50*a*, *L*_y_ = 35*a*, and *L*_z_ = 50*a*, where a is the Si lattice constant of 0.5431 nm. Periodic boundary conditions were applied in the *x*- and *z*-directions. The conical indenter was composed of a carbon diamond structure (C) with a lattice parameter of 0.3576 nm, and was located 1 nm above the Si substrate. To analyze the Si substrate’s response under the conical indenter, the indenter semi-angles (*α*) were chosen as 45°, 60°, and 70°.

The Si substrate consisted of fixed, thermostat, and Newtonian layers [12,18]. The fixed layer at the bottom of the substrate was treated as a rigid body for structural stability, and the thermostat layer above the fixed layer was used to ensure reasonable outward heat conduction away from the Newtonian layer through an NVT ensemble. The layer above the thermostat layer was the Newtonian layer, where the behavior of atoms was only governed by the forces between them, with no other constraints [19].

It is important to consider the interatomic potential to obtain accurate simulation results. Although many potential functions have been developed to describe the interaction between Si atoms, it has been demonstrated by many researchers that the Tersoff potential provides a reasonable deformation mechanism of Si [20]. In the present study, the Tersoff potential was used to compute the interaction between Si atoms (Si–Si) and diamond atoms (C–C) [21]. The interaction potential between the Si and C atoms was modeled using Morse potential [19].

Before the indentation procedure, the temperature of the Si substrate was equilibrated at either 1 K or 300 K. To equilibrate the temperature of the substrate, an NVE ensemble with a velocity rescaling method was used in the Newtonian layer [22,23]. Subsequently, the indentation load was applied by displacing the conical indenter into the substrate. The indentation speed was set to 50 m/s and the indentation depth was 3 nm. Retreating of the indenter was then conducted at the same speed. MD simulations were conducted using the open source MD program Large-Scale Atomic/Molecular Massively Parallel Simulator (LAMMPS) [24], and the atomistic configurations of the results were visualized using AtomViewer [25] and AtomEye [26].

## 3. Structural Phases of Si

Various structural phases of Si were observed during the nanoindentation simulation. The Si phases of Si-I, Si-II, BCT-5, Si-III (R8), and Si-XII (BC8) are schematically described in Figure 2 [27]. The Si-I phase is the initial diamond cubic structure with four-fold coordination, and Si-II is the tetragonal (*β*-tin) structure with six-fold coordination [2]. BCT-5 is the five-coordinated body-centered tetragonal structure that consists of four neighbors at a distance of 0.244 nm and one neighbor at 0.231 nm [28]. The Si-III (R8) and Si-XII (BC8) phases have four-fold coordination with similar structures [29].

## 4. Theoretical Method

### 4.1. Sneddon’s Solution

To investigate the deformation mechanism of the Si substrate during the loading procedure in more detail, the load of the indenter was compared with Sneddon’s solution [17], which is the relationship between a frictionless non-rigid cone and an elastic half plane. Sneddon’s solution is given by Equation (1).
(1)$$P=\frac{2}{\pi }E^{*}\tan\alpha \times h^{2},$$
where *P* is the load applied to the indenter, *α* is the semi-angle of the conical indenter, *h* is the indentation depth, and *E** is the reduced modulus. The reduced modulus was defined by Oliver and Pharr [16], as given in Equation (2).
(2)$$\frac{1}{E^{*}}=\frac{1-\nu_{s}^{2}}{E_{s}}+\frac{1-\nu_{i}^{2}}{E_{i}},$$
where *E* and *ν* are the elastic modulus and Poisson’s ratio, respectively. The subscripts *s* and *i* indicate the substrate and indenter, respectively. The elastic modulus and Poisson’s ratio for the substrate and indenter are *E_s_* = 130 GPa, *E_i_* = 1056.48 GPa, *ν_s_* = 0.3459, and *ν_i_* = 0.103, respectively [30,31].

### 4.2. Calculation of Mechanical Properties

In nanoindentation, hardness (H) is defined as follows [16]:(3)$$H=\frac{P_{\max}}{A_{c}},$$
where *P*_max_ is the maximum indentation load and A_c_ is the projected contact area. For the conical indenter, the projected contact area is expressed as follows:(4)$$A_{c}=\pi {\left(\tan\alpha \times h_{c}\right)}^{2},$$
where *α* is the semi-angle of the cone indenter and *h_c_* is the contact depth. In this study, we assumed that *h_c_* = *h* for the actual projected contact area [32,33].

The elastic modulus values of Si were calculated from the unloading line using Sneddon’s solution [17]. To measure the elastic modulus of the Si substrate, Equation (2) is modified to
(5)$$E_{s}=\left(1-\nu_{s}^{2}\right)\times {\left(\frac{1}{E^{*}}-\frac{1-\nu_{i}^{2}}{E_{i}}\right)}^{-1},$$
where *E* and *ν* are the elastic modulus and Poisson’s ratio, respectively, and the subscripts *s* and *i* indicate the substrate and indenter, respectively. *E** is the reduced modulus. According to Sneddon, the reduced modulus of the substrate is defined as given in Equation (6).
(6)$$E^{*}=\frac{1}{2h\tan\alpha }\frac{dP}{dh}=\frac{1}{2h\tan\alpha }S,$$
where *h* is the indentation depth, *α* is the semi-angle of the conical indenter, and the stiffness, *S*, is the slope of the unloading line. Typically, *S* is measured for the top one-third of the unloading line. In this study, *S* was measured in the range of 2.5 to 2.96 nm for the unloading line.

## 5. Results and Discussion

### 5.1. P-h Curves

To analyze the dependence of the Si substrate’s deformation mechanism on the indenter angle, the load–indentation depth (*P*-*h*) curves for indenter angles of 45°, 60°, and 70° were obtained, as shown in Figure 3. The black solid lines were obtained from Sneddon’s solution, corresponding to Equation (1). It is well known that if the indented material shows perfectly elastic deformation, then the unloading line falls on the loading line in the *P*-*h* curve [16]. In contrast, if plastic deformations largely occur in the indented material, the unloading line does not fall on the loading line.

In our simulation results, the discrepancies between the loading and unloading lines decreased with increasing indenter angles. The effect of temperature on the discrepancy could also be observed for the 60° and 70° indenter angles, whereas there was little difference between the 1 K and 300 K plots for the 45° indenter angle. In addition, the load curve for an indenter semi-angle of 45° was smaller than that obtained from Sneddon’s solution. In contrast, the load curves for the 60° and 70° indenter angles were similar and larger than those derived from Sneddon’s solution. It is worth noting that these discrepancies resulted from the frictionless assumption used in Sneddon’s solution [17,34,35]. Hence, it can be speculated that the fraction of plastic deformation, with respect to the total deformation under loading, decreased with the increasing indenter angle.

### 5.2. Loading Procedure

Cross-sectional views of the transformed region at an indentation depth of 3 nm are shown in Figure 4. Figure 4a,b illustrate the cases of the 45°, 60°, and 70° indenter semi-angles at 1 K and 300 K, respectively. To identify the Si phases, the original Si phase atoms (Si-I) were not captured, while the transformed Si atoms were depicted using three colors; the blue, red, and green atoms indicate the Si-II, BCT-5, and Si-III and Si-XII phases, respectively.

As shown in Figure 4, the distributions of the transformed Si phases were quite different depending on the indenter angle. When the Si substrate was indented by the 45° indenter, the structure of the Si atoms at the interfacial area between the indenter and the substrate was transformed from diamond to the Si-II and BCT-5 phase structures. However, the number of transformations to the Si-II and BCT-5 phases at the interfacial area decreased with increasing indenter angles. These transformed phase distributions are associated with pressure concentration [1,36,37]. To monitor the pressure distribution induced by the conical indenter, the (001) plane cross-sectional views of the Si substrate at a temperature of 1 K were captured and are presented in Figure 5. Here, the color of the atoms indicates the level of hydrostatic pressure. The pressure distributions depicted in Figure 5 matched well with the phase distributions of the Si atoms shown in Figure 4a.

To analyze the detailed behavior of the phase transformations, the ratio of the Si-II and BCT-5 atoms (*N*_Si-II and BCT-5_) to the total transformed atoms (*N*_Total_) was calculated (Table 1). It has been previously reported [13,18] that the Si-II and BCT-5 phases are related to plastic deformation, while metastable Si phases, such as Si-III and Si-XII, are associated with elastic deformation. This trend was also observed in our simulation results, as shown in Figure 3 and Figure 4. For example, the transformed atoms in the case of the 70° indenter at 1 K were almost the same as those of the Si-III and Si-XII phases (Figure 4(a3)). However, at a temperature of 300 K (Figure 4(b3)), a large number of transformations to the Si-II and BCT-5 phases occurred at the subsurface. This trend was evident in the *P*-*h* curves shown in Figure 3c. Therefore, the ratio of the Si-II and BCT-5 atoms to the total number of transformed atoms (*N*_Si-II and BCT-5_/*N*_Total_) can be used as the ratio of plastic deformation to total deformation.

In addition, it can be seen that the ratio of Si-II and BCT-5 atoms at 300 K is larger than that at 1 K, implying that more plastic deformation occurs at 300 K than at 1 K. These results can be explained by the relationship between the temperature and pressure required for phase transformation. Khayyat et al. [38] utilized a Vickers diamond indenter (equivalent cone semi-angle of 70.3°) and experimentally showed that the pressure required to cause transformation increases with decreasing temperature. As shown in Table 1, this effect of temperature on the ratio was observed for the 60° and 70° indenters, whereas only a small discrepancy between the ratios at 1 K and 300 K was observed for the 45° indenter angle. This indicates that while the influence of temperature on plastic deformation is significant for the 60° and 70° indenters, it is not as significant for the 45° indenter angle, which is consistent with the *P*-*h* curve results.

The phase transformation in the Si substrate during indenter loading is shown in Figure 6. Figure 6a–c illustrate the 60° indenter case at the indentation depths of 1 nm, 2 nm, and 3 nm, respectively. As the hydrostatic pressure beneath the indenter tip increases with loading (Figure 5b), the area of phase transformations to Si-II, BCT-5, Si-III, and Si-XII structures increases in the substrate. Specifically, the number of Si-II and BCT-5-phase atoms (related to plastic deformation) increases beneath the indenter tip, where the location matches the large hydrostatic pressure zone. The results indicate that the phase transformation behavior of the Si structure is dependent on the hydrostatic pressure distribution.

### 5.3. Unloading Procedure

To analyze the deformation mechanism of the substrate under unloading conditions, the ratio of the final depth (*h_f_*) to the maximum indentation depth (*h*_max_) was plotted as a function of the indenter angle (Figure 7) [38]. The final depth was defined as the point at which *P* = 0 in the unloading line. In the range 0 ≤ *h_f_*/*h*_max_ ≤ 1, the lower bound corresponds to the case of perfect elastic recovery, while the upper bound indicates the fully plastic deformation case. As displayed in Figure 7, the ratio increased with decreasing indenter angle, and the ratio at 300 K was larger than that at 1 K. These results indicate that at a temperature of 1 K, the Si substrate indented by a blunt indenter recovers in a more elastic manner, and elastic recovery is greater. In particular, the ratio for the 70° indenter angle at 1 K was 0.143, which indicates that almost perfect elastic recovery occurs in the Si substrate.

The distribution of residual transformed atoms in the substrate after unloading is shown in Figure 8. Figure 8a,b illustrate the cases of indenter angles of 45°, 60°, and 70° at temperatures of 1 K and 300 K, respectively. From a comparison with Figure 4, it can be seen that most of the Si-III and Si-XII atoms disappeared, and the amorphous-phase Si comprising Si-II, BCT-5, Si-III, and Si-XII atoms was formed in the substrate. In addition, it can be observed that the shape of amorphous Si was similar to the distribution of Si-II and BCT-5 atoms at an indentation depth of 3 nm. For instance, for the 45° indenter angle, cone-shaped amorphous Si was formed in the substrate, which was similar to the distribution of the Si-II and BCT-5 atoms at an indentation depth of 3 nm, as shown in Figure 4. In contrast, for the 70° indenter angle at 1 K, most of the transformed atoms returned to the original Si structure and only a few transformed atoms remained at the surface, as shown in Figure 8(a3).

To study the ratio of elastic recovery under unloading, the ratio of atoms that returned to their original structure upon unloading to the total number of transformed atoms at an indentation depth of 3 nm, (*N*_Returned atom_/*N*_Total_), was calculated (Table 2). From the results given in Table 2, it is evident that the ratio *N*_Returned atom_/*N*_Total_ increased with increasing indenter angle and the ratio at 1 K was larger than that at 300 K, which indicates that the elastic recovery occurred more readily under blunter indenter unloading and at lower temperatures. These atomistic configuration results are consistent with the *P*-*h* curve trend shown in Figure 2.

Figure 9 shows the phase transformation in the Si substrate during the unloading procedure. Figure 9a–c illustrate the 60° indenter case at the indentation depths of 2 nm, 1 nm, and 0 nm, respectively. During indenter unloading, most of the Si-III- and Si-XII-phase atoms disappear, while the Si-II- and BCT-5-phase atoms remain in the Si substrate. The phase transformation behavior under the unloading procedure describes that the Si-III and Si-XII phases are associated with the elastic deformation of Si, whereas Si-II and BCT-5 phases are related to the plastic deformation of Si [18,22].

### 5.4. Mechanical Properties

The hardness (H) of the Si substrate was presented as a function of the indentation depth (*h*) for each indenter angle at 300 K, and the results are shown in Figure 10. The hardness was calculated from the *P*-*h* curve using Equations (3) and (4). For the 45° and 60° indenters, it was observed that the hardness decreased with increasing indentation depth, and then converged to a constant value. However, for the 70° indenter, the hardness remained almost constant during loading. This trend is also demonstrated by the indentation size effect [32,39,40], which appeared strongly in the case of the sharper indenter because of the relatively small penetration volume.

From Figure 10, the convergence values of hardness were obtained as 20.1, 14.21, and 11.36 GPa for the indenter semi-angles of 45°, 60°, and 70°, respectively, which are the average values at an indentation depth ranging from 2.9 to 3 nm. Experimental values of the hardness of single-crystal Si have been reported to be 12–14 GPa at 300 K [41]. Thus, the result of the 60° indenter was found to be in good agreement with experimental values. In addition, it can be seen that the obtained hardness decreases with increasing indenter angle. This angle-dependent hardness is associated with plastic flow. According to Prasad et al. [42], the hardness values decreased with an increase in the elastic deformation zone surrounding the plastic zone. As shown in Figure 4b, the size of the elastic zone surrounding the plastic zone increases with increasing indenter angle. Hence, it is reasonable to conclude that the hardness of Si depends on the indenter angle at the atomic scale.

The elastic modulus of the Si substrate was calculated for each indenter angle at 300 K using Equations (5) and (6), and the results are presented in Figure 11. The experimental value of the elastic modulus of Si in the (010) plane was reported to be 130 GPa [29,43]. As shown in Figure 9, although the obtained elastic modulus decreased with increasing indenter angle, the values were within a reasonable range when compared to experimental values. When *α* = 45°, the difference between the obtained result and the experimental value was 5.3%, while for *α* = 60° and 70°, the differences were 2.9% and 6.5%, respectively. Therefore, it can be concluded that the results obtained for the indenter with a 60° semi-angle were best fitted to the experimental values.

## 6. Conclusions

This research revealed atomistic insights into the effect of indenter shape on the phase transformation behavior of a Si substrate via MD simulations. Upon indenter loading, phase transformations to Si-II, BCT-5, Si-III, and Si-XII were observed beneath the indenter tip. For the indenter unloading procedure, the number of Si-III and Si-XII atoms significantly decreased, but the Si-II- and BCT-5-phase atoms remained in the Si substrate. The results exhibited that the Si-III and Si-XII phases describe elastic recovery, while the Si-II and BCT-5 phases describe plastic deformation. Among the three conical indenters with semi-angles of 45°, 60°, and 70°, the 70° indenter case showed greater elastic recovery, which was elucidated based on the *P*-*h* curve and atomistic deformation behaviors. Furthermore, the mechanical properties of hardness and elastic modulus decreased with increasing indenter angle, and the 60° indenter case showed good agreement with previous experiments. The present findings obtained from MD simulations are expected to give useful insights into the deformation mechanism of thin-film Si under nanoindentation testing.

## Figures and Tables

**Figure 1 nanomaterials-12-02071-f001:**
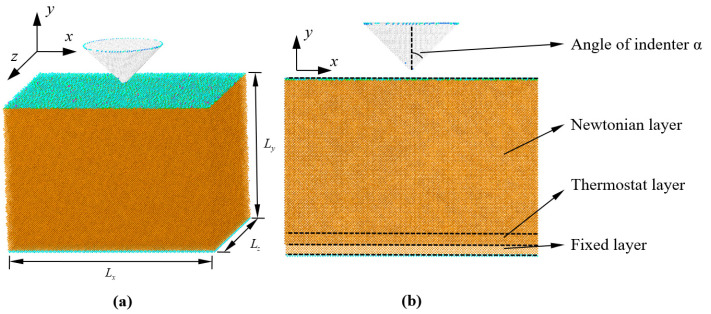
Computational model of a Si single-crystal substrate with a conical indenter: (**a**) three-dimensional model; (**b**) schematic diagram of the Si substrate and conical indenter.

**Figure 2 nanomaterials-12-02071-f002:**
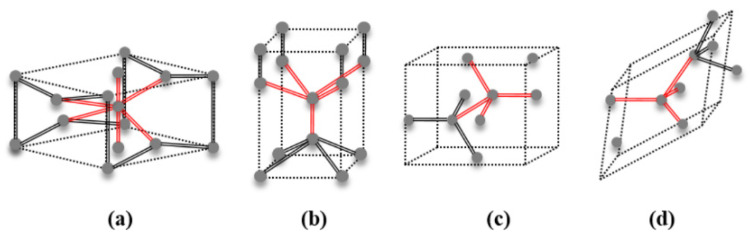
Crystallographic structures of the Si phases observed during nanoindentation: (**a**) Si-II, a tetragonal (*β*-tin) structure with six-fold coordination; (**b**) BCT-5, a body-centered tetragonal structure with five-fold coordination; (**c**) Si-III (BC8), a cubic structure with four-fold coordination; and (**d**) Si-XII (R8), a rhombohedral distortion structure with four-fold coordination.

**Figure 3 nanomaterials-12-02071-f003:**
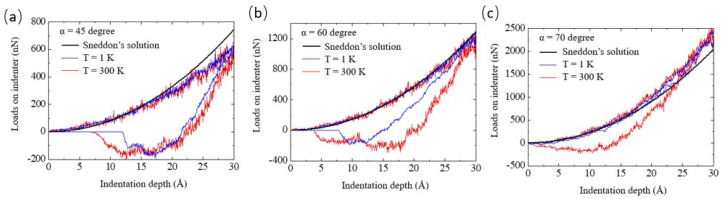
Load−indentation depth curves at temperatures of 1 and 300 K for indenter angles (*α*) of (**a**) 45°, (**b**) 60°, and (**c**) 70°.

**Figure 4 nanomaterials-12-02071-f004:**
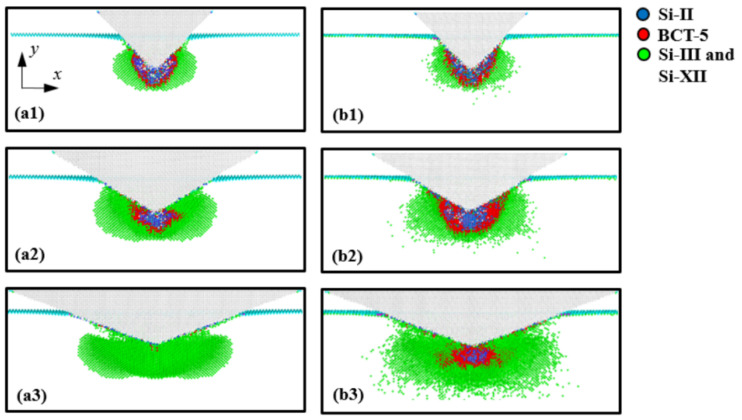
Cross-sectional views of the (001) plane of the transformed Si substrate at an indentation depth of *h* = 3 nm. Images (**a1**–**a3**) correspond to indenter angles of 45°, 60°, and 70° at a temperature of 1 K, and (**b1**–**b3**) correspond to indenter angles of 45°, 60°, and 70° at 300 K, respectively.

**Figure 5 nanomaterials-12-02071-f005:**
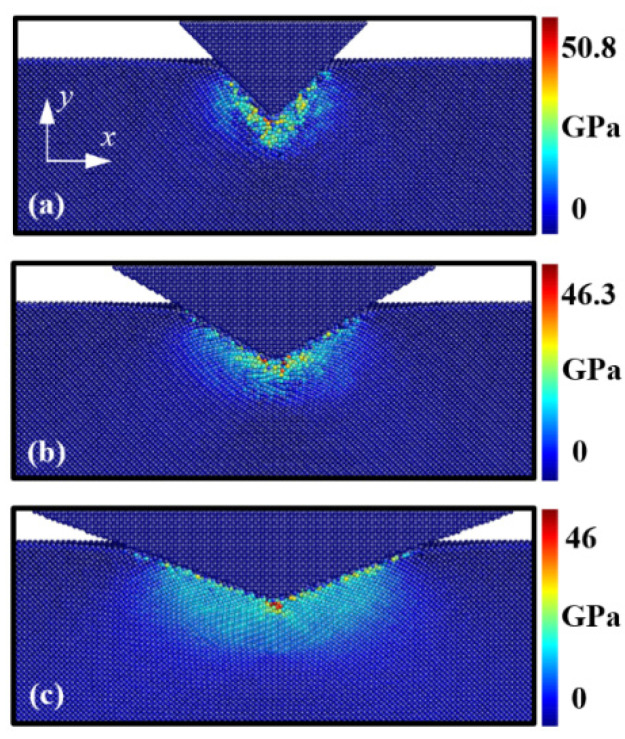
Cross-sectional views of the (001) plane of the hydrostatic pressure distribution in the Si substrate at an indentation depth of *h* = 3 nm. Images (**a**–**c**) correspond to indenter angles of 45°, 60°, and 70° at a temperature of 1 K, respectively.

**Figure 6 nanomaterials-12-02071-f006:**
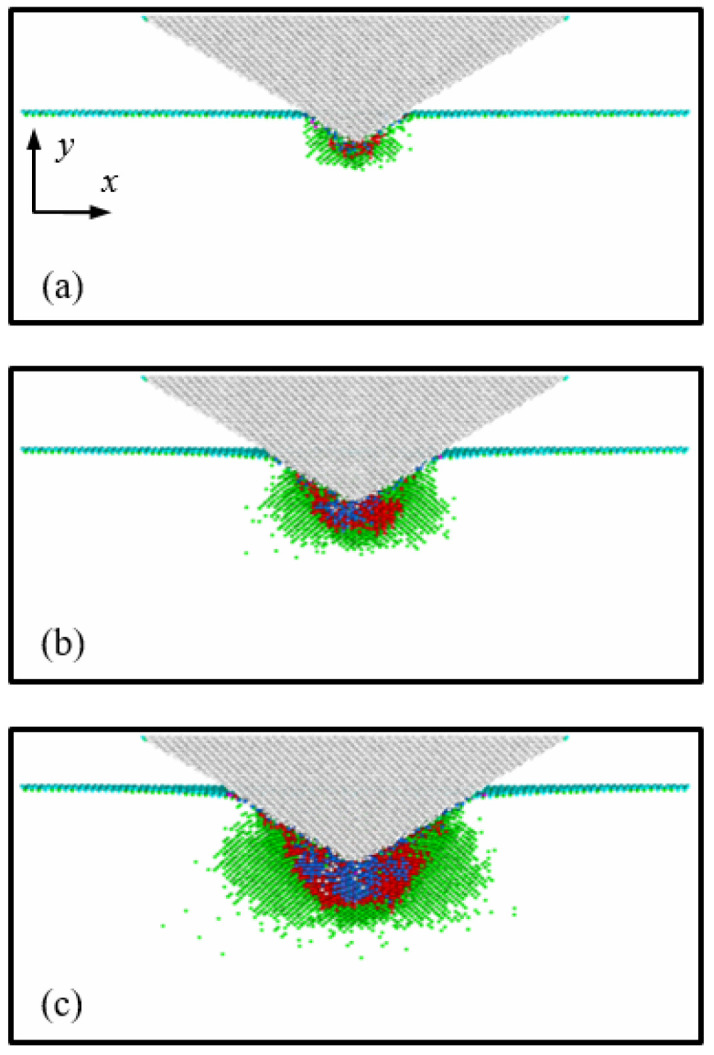
Cross-sectional views of the (001) plane of the transformed Si substrate during indenter loading. Images (**a**–**c**) correspond to the 60° indenter angle case at the indentation depths of 1 nm, 2 nm, and 3 nm, respectively.

**Figure 7 nanomaterials-12-02071-f007:**
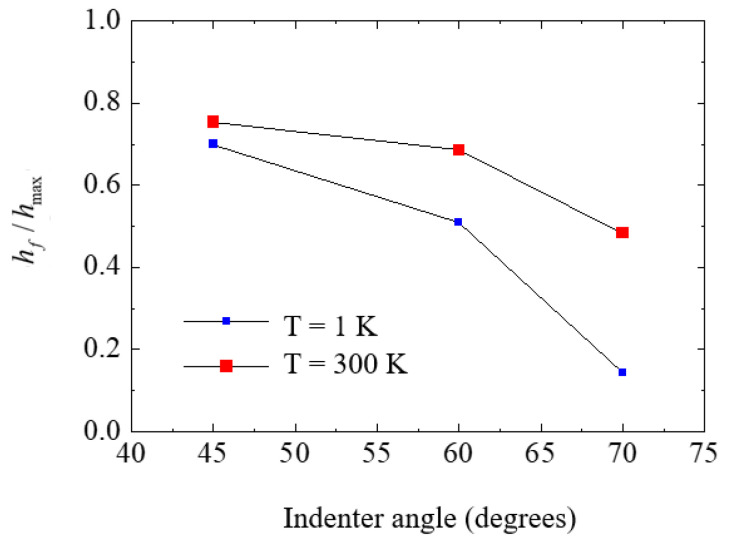
Ratio of the final depth, *h_f_*, to the maximum indentation depth, *h*_max_, versus the indenter angle *α*.

**Figure 8 nanomaterials-12-02071-f008:**
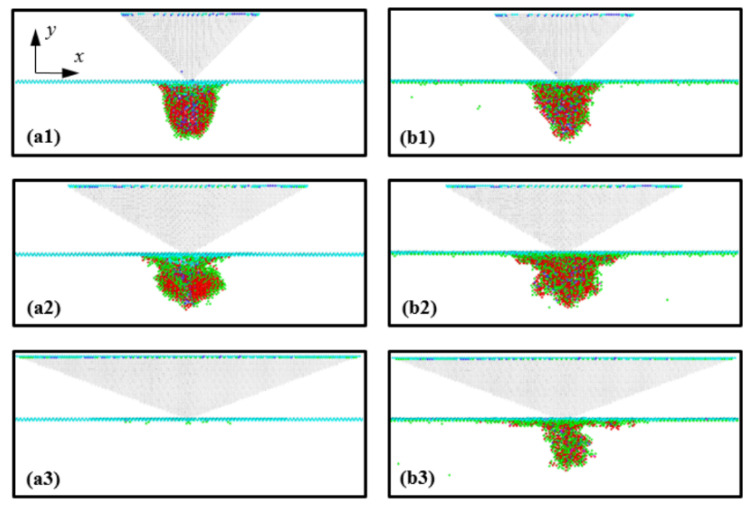
Views of the residual transformed atoms in the (001) plane of the Si substrate after unloading. Images (**a1**–**a3**) correspond to indenter angles of 45°, 60°, and 70° at a temperature of 1 K, and (**b1**–**b3**) correspond to indenter angles of 45°, 60°, and 70° at 300 K, respectively.

**Figure 9 nanomaterials-12-02071-f009:**
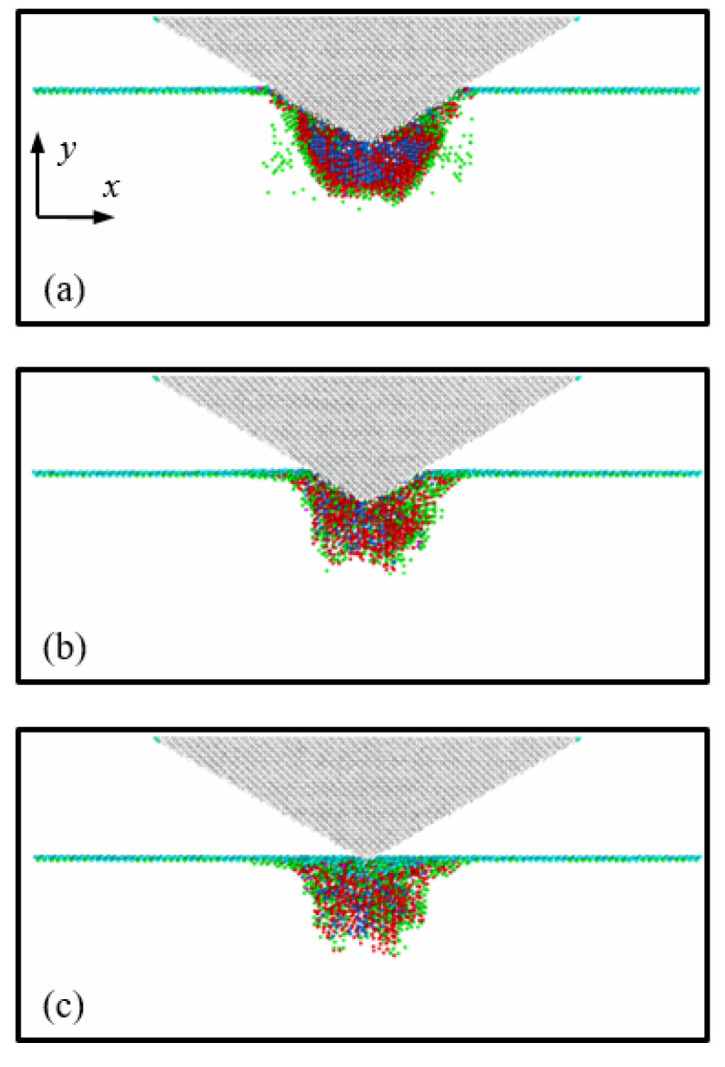
Cross-sectional views of the (001) plane of the transformed Si substrate during indenter unloading. Images (**a**–**c**) correspond to the 60° indenter angle case at the indentation depths of 2 nm, 1 nm, and 0 nm, respectively.

**Figure 10 nanomaterials-12-02071-f010:**
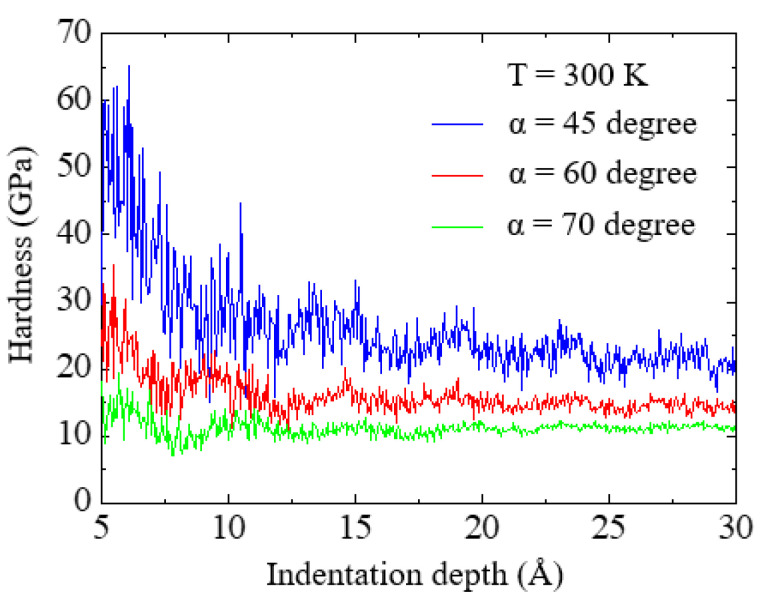
Hardness–indentation depth curves at a temperature of 300 K for the three indenter angles.

**Figure 11 nanomaterials-12-02071-f011:**
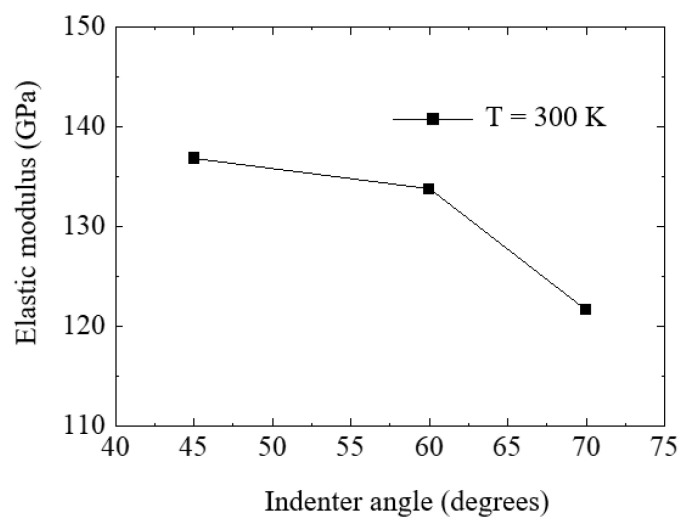
Computed elastic modulus values at a temperature of 300 K for the three indenter angles.

**Table 1 nanomaterials-12-02071-t001:** Ratio of Si-II and BCT-5 atoms (*N*_Si-II and BCT-5_) to the total number of transformed atoms at the maximum indentation depth (*N*_Total_).

Indenter Angle	*N*_Si-II and BCT-5_/*N*_Total_	
	1 K	300 K
45°	0.34	0.363
60°	0.162	0.362
70°	0.032	0.164

**Table 2 nanomaterials-12-02071-t002:** Ratio of atoms that returned to the original structure upon unloading (*N*_Returned atom_) to the total number of transformed atoms at the maximum indentation depth (*N*_Total_).

Indenter Angle	*N*_Returned atom_/*N*_Total_	
	1 K	300 K
45°	0.546	0.47
60°	0.783	0.736
70°	0.955	0.877

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
