# Peer review of "Atomistic Insights into the Phase Transformation of Single-Crystal Silicon during Nanoindentation"

_nanomaterials, 2022, doi:10.3390/nano12122071_

Round 1
Reviewer 1 Report
The manuscript by Chung et al. reports the structural phase transformation during nanoindentation using simulation tools. The following points should be addressed before publication:
1. The manuscript title highlights the atomistic insights, but the illustration of the real atomic structures and bond lengths/angle/symmetry changes during the phase transformation via nanoindentation is insufficient. Figure 2 is a good effort; however, this figure is too general. Please provide a separate figure illustrating more detailed atomic level information such as the changes of atomic position/bond lengths/angles during the phase transition/nanoindentation.
2. The references are too old; please include and discuss more recent findings (papers published in the recent three years) in this field.
Reviewer 2 Report
In this work, Chung et al. present their atomistic molecular dynamics simulations on nano-indentation of Si crystals. They observed phase transformation at an atomic level during the indentation. The atomistic code LAMMPS has been well-established and widely used in simulation of many processes. The Tersoff and the Mose potentials have been for long employed for describe Si-Si and C-C, and Si-C interactions, respectively. The obtained information is helpful to people to obtain insight into the mechanism of nano-dentation. The manuscript is written properly. The text is in the scope of this Journal. I am glad to propose acceptance of this paper for publication in Nanomaterials.
Reviewer 3 Report
Nanomaterials:
Atomistic insights into the phase transformation of single crystal silicon during nanoindentation
Comments:
1. Add abbreviations separately in the article!
2. The conclusions need to be redrafted / written differently - so that the contents of the abstract are not repeated.
